# Effect of a hybrid team-based advanced cardiopulmonary life support simulation program for clinical nurses

Hye Won Jeong[1,2], Deok Ju[1], Ae Kyong Lee[1], Jung A Lee[1], Na Ru Kang[1], Eun Jeong Choi[1], Shin Hye Ahn[1,2], Sun-Hee Moon[2,3]*

1 Department of Nursing, Chonnam National University Hospital, Gwangju, South Korea, 2 College of Nursing, Chonnam National University, Gwangju, South Korea, 3 Chonnam Advanced Reality-based Education Center, Gwangju, South Korea

* sunnymon@jnu.ac.kr

**Data Availability Statement:** We uploaded the raw data to 'protocols.io'; https://protocols.io/file/jtf77qze.sav.

## Abstract

Background: During in-hospital cardiac arrest events, clinical nurses are often the first responders; therefore, nurses require sufficient advanced cardiac life support (ACLS) competency. This study aimed to verify the effects of a hybrid team-based ACLS simulation (HTAS) program (developed in this study) on nurses' ACLS performance, specifically ACLS knowledge, cardiopulmonary resuscitation (CPR) self-efficacy, and CPR-related stress. Methods: The developed HTAS comprised four lecture videos, one team-based skills training video, and a team-based ACLS simulation. A quasi-experimental pretest-posttest design with a comparison group (CG) was used to evaluate the effectiveness of the HTAS. Of the 226 general ward nurses with more than 6 months of clinical experience, 117 were allocated to the intervention group (IG), which attended the HTAS, and 109 to the CG, which attended only basic ACLS training. Results: The IG's ACLS performance significantly improved (t = 50.8, $p < 0.001$) after the training. Relative to the respective pretest conditions, posttest ACLS knowledge (t = 6.92, $p < 0.001$) and CPR self-efficacy (t = 6.97, $p < 0.001$) of the IG also significantly increased. However, when the mean difference values were compared, there was no significant difference between the two groups with respect to ACLS knowledge (t = 1.52, $p = 0.130$), CPR self-efficacy (t = -0.42, $p = 0.673$), and CPR stress (t = -0.88, $p = 0.378$). Conclusion: The HTAS for ward nurses was effective at enhancing the nurses' ACLS performance. It is necessary to develop effective training methods for team-based ACLS and verify the sustained effects of such training.

## Introduction

The global incidence of in-hospital cardiac arrest (IHCA) has not been investigated; however, in the United States, an estimated 9–10 events of cardiac arrest per 1000 admissions occurred from 2008 to 2017 [1], while in Korea, 3.71 events per 1000 admissions were estimated to have occurred from 2003 to 2013 [2]. Although cardiac arrest is associated with high mortality

**Funding:** -The 10th authors of this study, Sun-hee Moon, received the fund. -This work was supported by the National Research Foundation of Korea (NRF) grant funded by the Korea government (MSIT) (No. NRF-2020R1I1A3063639). There was no additional funding for this study.

**Competing interests:** The authors have declared that no competing interests exist.

rates, survival rate can be increased by 2.15 odds if bystanders perform cardiopulmonary resuscitation (CPR) [3]. When cardiac arrest events occur in hospitals, nurses are often the first responders, and they provide patient care throughout the first 24 hours after IHCA [4]. Therefore, nurses' competence during the initial assessment and performance of CPR is crucial [4]. To safeguard this competence, CPR training is deemed a key component in nurses' management of life-threatening crises in clinical settings.

Among the educational methods used to instill knowledge retention and CPR skills, simulation is considered the most effective [5, 6]. The use of simulation education for healthcare providers significantly improves clinical skills [7], positively affects patient safety (partly by reducing medical errors), does not threaten patient safety, and can provide a safe learning environment [8]. In particular, a team-based approach in simulation education has been emphasized because healthcare providers—including doctors, nurses, and emergency medical technicians—can form effective teams that implement advanced cardiac life support (ACLS) during IHCA events [6, 9]. Such team-based ACLS simulation training has been reported to improve communication, collaboration, teamwork, and leadership [10]. The ACLS simulation training program developed by the American Heart Association (AHA) emphasizes the effectiveness of teamwork among healthcare providers in providing medical treatment, as well as achieving communication skills and collaboration [11]. The development of CPR simulation-based education for clinical nurses should focus on improving team performance through a team-based approach.

In CPR education, low-dose, high-frequency CPR training is effective for skill retention [12]. Extant literature has discussed various methods to improve CPR skill retention. Online CPR training has been recommended as an effective alternative for facilitating skill retention because access to such training is not bound by time or location [13]. Recently, with the spread of COVID-19, the importance of distance learning in CPR training has increased [14]. According to review studies, video instruction and online training methods improve chest compression performance in terms of rate, depth, efficacy, and knowledge [14, 15]. Mixing these various teaching media with simulation could enhance the effectiveness of team-based ACLS training. Hybrid CPR training includes a mixture of non-face-to-face methods using video or computer-based course material, along with face-to-face instructor-led training using CPR simulation [16]. A review of the effectiveness of alternative CPR training methods found hybrid methods to be more effective at facilitating knowledge retention and performance compared to standard instructor-led training [16]. Therefore, to improve the CPR competency of clinical nurses, an effective educational program might consist of a hybrid design that delivers ALCS knowledge and skills in an iterative and convenient way, allowing trainees to practice the acquired knowledge and skills on high-fidelity manikins.

The outcomes of CPR training include the performance domain, consisting of elements such as chest compression rate and depth, hands-off time, and operation of defibrillators and bag-valve masks [5, 15]. The knowledge domain, on the other hand, comprises elements such as knowledge of the ACLS algorithm, electrocardiogram (ECG) analysis, and drug administration [5, 15, 17]. Finally, the attitudes domain consists of concepts such as self-efficacy and confidence [15]. A study of the effectiveness of team-based CPR simulation training found that such training enhanced satisfaction, attitudes, perceptions, and acquisition of knowledge and skills [9]. The effectiveness of team-based CPR simulation should be comprehensively evaluated based on important factors in each area of knowledge, skills, and attitudes.

The AHA publishes CPR guidelines every 5 years [18], and most CPR simulation training programs are based on these guidelines. In previous studies [19, 20], training programs based on the Korean advanced life support (KALS) curriculum—a standard CPR curriculum established by the AHA for the Korean context—have been conducted face to face using manikins

for 4.5–6 hours. To increase the effectiveness of team-based CPR simulations, an online program that can conveniently and repeatedly deliver the ACLS guidelines can be combined with simulation training. A team-based non-face-to-face CPR education model rooted on KALS guidelines could provide ACLS education more conveniently and maintain its effectiveness. Thus, this study aimed to develop a hybrid team-based ACLS simulation (HTAS) training program mixed with online training content and to validate the effect of this training on nurses' performance, knowledge, and self-efficacy. This study's hypotheses were that, compared with nurses undergoing a basic online ACLS program, nurses in the HTAS intervention group would have (1) superior ACLS performance (primary outcome) and (2) superior ACLS knowledge and CPR self-efficacy (secondary outcomes).

## Materials and methods

### Theoretical framework

The HTAS framework was based on the National League for Nursing (NLN) Jefferies Simulation Theory [21] (Fig 1), wherein context, background, design, simulation experience, facilitator, educational strategy, participant, and outcome are the core elements of simulation education. Based on the theory, this study considered pertinent situations and background factors and thus developed simulation programs based on hospital clinical sites and identified the impact of general characteristics on simulation education results. The facilitator variable was developed using standardized scenario development and evaluation guidelines. The HTAS program included two pieces of online content and one offline team-based simulation exercise. Furthermore, ACLS knowledge, CPR self-efficacy, CPR stress, and team ACLS performance were measured as the outcome variables of the team-based ACLS simulation program to confirm its effectiveness.

### Development of the HTAS

The preliminary simulation scenario was prepared based on CPR situation problem lists generated by 10 rapid response team nurses from the hospital where the intervention was to be administered. The problem lists noted issues that arise during ward CPR, including inadequate intubation support, delayed suction during intubation, lack of knowledge about emergency medication, inexperience with defibrillator use, delayed preparation of intravenous lines, and poor understanding of team-based CPR processes. Consequently, the scenario was constructed with a focus on these deficiencies. Since problems related to airway management were frequently reported during ACLS, the endotracheal intubation process and assistance was particularly emphasized in this study.

A research team was formed to develop the HTAS framework (Fig 2). The research team comprised five nurses with more than 12 years of clinical experience and one emergency nursing professor. The HTAS was developed based on the KALS algorithm [20] and consisted of three phases. The first phase comprised four lectures on basic ACLS offered as online videos. They covered the KALS algorithm, rapid defibrillation, ACLS medication, and advanced airway management. In the second stage, a real patient scenario of chronic kidney disease with hyperkalemia was introduced for team-based skills training, along with an ACLS video demonstration on role establishment, ECG reading, and communication. A debriefing about the demonstration was also developed. The third stage entailed a pre-briefing session, which included role orientation, scenario explanation, simulation room, and simulator introduction for 20 minutes. A team-based ACLS simulation involving six members (three nurse roles, two physician roles, and one emergency medical technician role) ran for 20 minutes using a high-fidelity simulator (Nursing Anne Simulator, Laerdal Medical, Stavanger, Norway). The

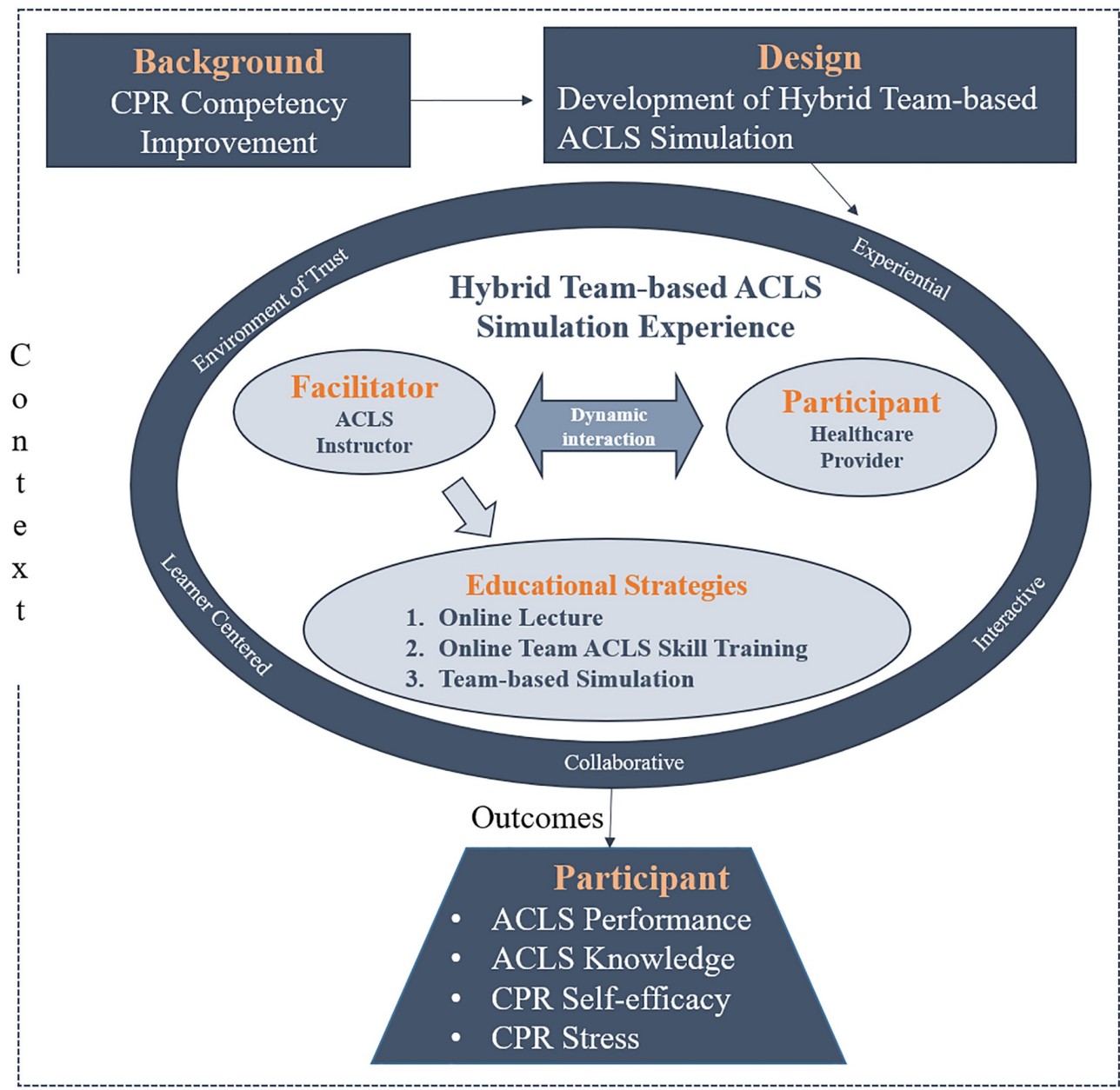

**Fig 1. Conceptual framework of this study.**

simulation process included an initial assessment, pulse check, team activation, chest compressions, ECG reading, defibrillation, medication administration, and endotracheal intubation assistance. The debriefing encouraged reflective thinking through personal contemplation, and the team debriefing included reflecting on the simulation situation with the faculty for 40 minutes.

## Verification of the HTAS

**Study design, setting, and sample size.** This study used a pretest-posttest quasi-experimental design to evaluate the effectiveness of the HTAS for training clinical nurses. This study

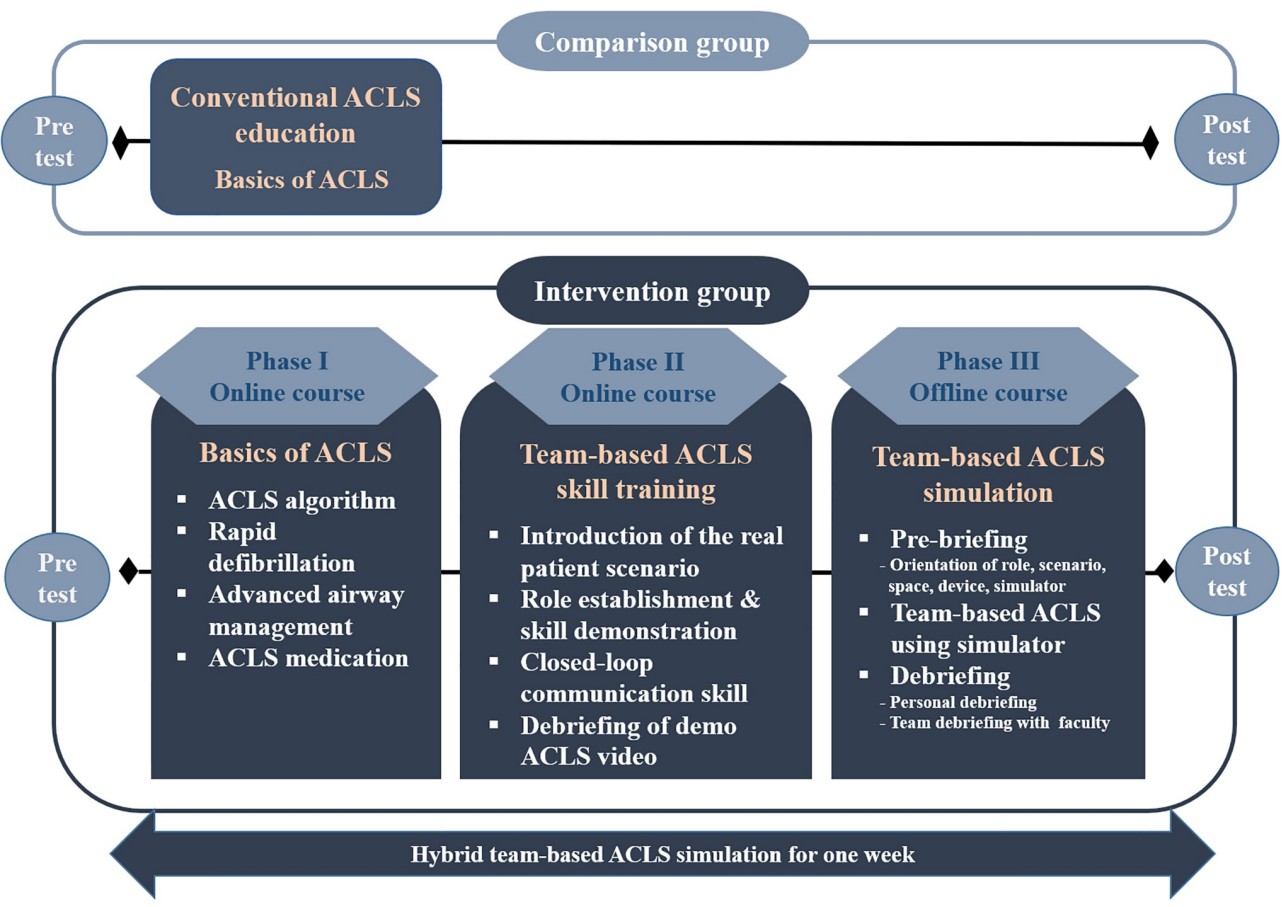

**Fig 2. Research design and simulation program in this study.**

was carried out at an 1100-bed tertiary hospital C in G metropolitan city, South Korea, in August and September 2021. During the study period, the hospital consisted of 19 medical and surgical wards, and a total of 17 wards participated, except for two designated as COVID-19 wards. All medical staff in this hospital were mandated to undergo CPR training once every 2 years, and if they participated in this study, it was recognized as one CPR training.

The sample size determination was based on the effect size used in a previous study that performed ACLS simulation education for clinical nurses [22]. The minimum number of participants needed to maintain the significance level ($\alpha$ = 0.05), effect size (f = 0.05), and power (1-ß = 0.95) was determined to be 210 using G*Power 3.1.9.4 software (Heinrich Heine University Düsseldorf, Düsseldorf, Germany). A 15% dropout rate was assumed, so the required sample size was increased to 241 participants: 121 participants in the intervention group (IG) and 120 participants in the comparison group (CG).

**Participants.** The participants were nurses who had worked in a general ward for more than 6 months and were recruited via convenience sampling. The reason for limiting the clinical experience of nurses to over 6 months was that, prior to this duration, nurses would not yet have adapted to their work routines [23], making it inappropriate to evaluate the educational effects of the HTAS in the presence of deficit baseline competency. The exclusion criteria were (1) nurses who worked in the pediatric ward, intensive care unit, or emergency department; (2) nurses with valid basic life support (BLS) or ACLS certificates; and (3) nurses who did not

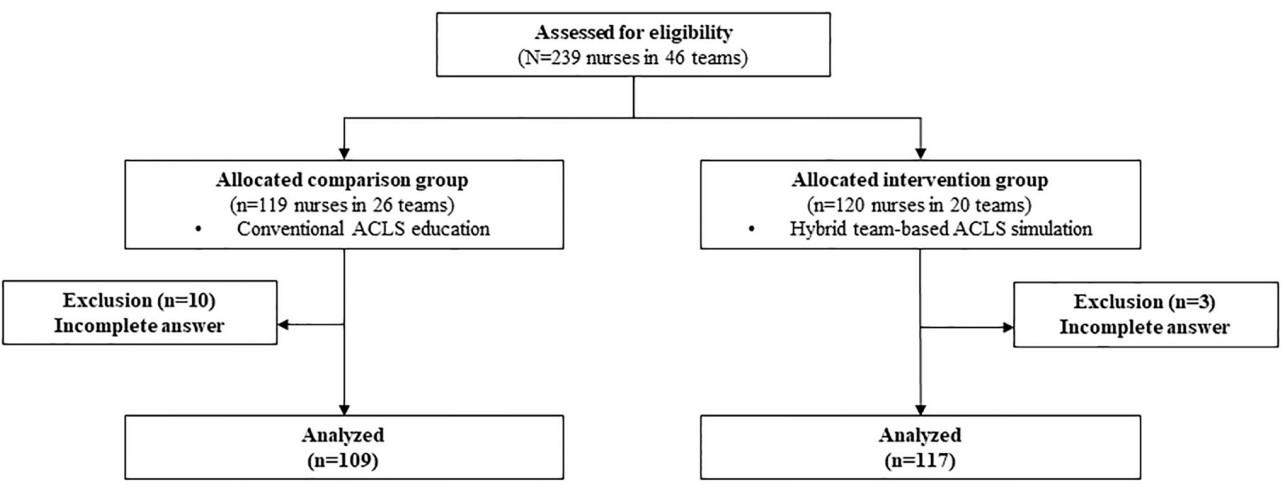

**Fig 3. Participant flow diagram.**

respond to the questionnaire. Nurses who worked in the intensive care unit or emergency department were excluded because it was difficult to measure the effectiveness of the HTAS if ACLS was performed during the training period, as cardiac arrests occur frequently. Since the HTAS developed in this study was constructed based on adult ACLS, the pediatric ward was also excluded.

An educational notice was uploaded to the hospital website. According to the selection criteria, 239 eligible participants were recruited (Fig 3). For ACLS performance measurements, groups of five to six nurses were organized to form a total of 46 teams. o prevent treatment diffusion, participants were first assigned to the CG group in the order in which they expressed their intention to participate by SMS. After the CG's conventional education was completed, the IG's treatment was conducted. Ten participants in the CG and three participants in the IG were excluded due to incomplete answers.

**Intervention.** Participants in the IG underwent the HTAS for 1 week. Participants accessed hospital C's online learning system and underwent basic ACLS (four lectures, 20 min 40 s) and team-based skills training (one lecture, 25 min 17 s) developed by the research team (S1 Fig). The participants' first completion of the training was recorded in the online learning system, and the participants could review the contents as they desired. On the last day of the intervention, the IG participated in the team-based CPR simulation instructed by the research team (pre-briefing, simulation, and debriefing, 80 min). Since the team-based ACLS simulation of the HTAS program was developed for six members, the researcher played the role of team member and facilitator when there was a shortage of team members.

**Comparison.** Participants in the CG accessed hospital C's online learning system and underwent basic ACLS training for 1 week. The participants' initial completion of the training was recorded in the online learning system, and the CG participants could also review the contents as they desired.

**Outcomes.** The participants responded to the ACLS knowledge, CPR self-efficacy, and CPR stress online pre-questionnaire using a Google Forms survey. After the self-administered questionnaire was completed, the participants underwent the respective training programs designed for the IG and CG.

ACLS performance was assessed by evaluators from the research team. To establish concordance between two evaluators using five ACLS simulation videos, first, the two evaluators

clearly agreed on the meaning of each item in the ACLS performance checklist. Second, after watching five ACLS simulation videos, two evaluators independently produced ACLS performance scores based on the checklist. The concordance between the scores of the two evaluators was a weighted kappa of 0.83. Third, when the two evaluators produced different performance scores after watching the ACLS simulation video, the reasons were discussed until a consensus was reached. After inter-rater agreement was reached, the ACLS performance was assessed. To evaluate the participants' ACLS performance, the evaluators assessed behavior with a checklist while watching the simulation in the same room with the participants.

After the ACLS performance assessment, the participants responded to the post-training questionnaire, which consisted of the same items as the pre-training questionnaire. The tools used in the evaluation of this study are described in the following subsections.

*ACLS performance.* The ACLS performance tool was developed based on the 2020 KALS [19] and 2020 AHA CPR [18] guidelines. The content validity of the tool was verified by the five experts who participated in the ACLS knowledge content validation. The content validity index (CVI) for each item of this tool was 0.9–1.0, and the total CVI was 0.992. The expressions of the five items were modified based on the opinions of experts. Additionally, to clarify the scoring criteria for each item, two researchers watched five ACLS simulation videos and scored them using the developed ACLS performance tool. The weighted kappa value of the two researchers was obtained to evaluate inter-measurement consistency, and it was found to be very good at 0.83 [24]. This tool consists of 40 items, including BLS performance (11 items), CPR team activation (6 items), ACLS performance (9 items), endotracheal intubation assistance and suction (7 items), and teamwork (7 items). These were scored on a 4-point Likert scale ranging from 1 (very poor) to 4 (very good), with higher scores indicating better ACLS performance. This tool's Cronbach's alpha was 0.99.

*ACLS knowledge.* The ACLS knowledge tool was developed based on the 2020 KALS [19] and 2020 AHA CPR [18] guidelines. As per the literature, 20 preliminary items were developed, and five experts (one nursing professor, one emergency medicine professor, and three nurses with master's degrees and over 15 years of clinical experience) were recruited to verify the items' content validity. The CVI of each item of this tool was 0.8–1.0, and the total CVI was 0.985. The tool consists of 20 items in total, including circulation check, help request, chest compressions, airway maintenance and artificial respiration, ECG analysis, emergency medication, defibrillation, and advanced airway intubation. The tool's scores range from 0–20, with higher scores indicating better ACLS knowledge.

*CPR self-efficacy.* The CPR self-efficacy tool was developed by Schlessel, Rappa [25], and it was later modified and supplemented by Byun, Park [26]. This tool consists of 12 items measured on a 10-point Likert scale ranging from 1 (not confident at all) to 10 (very confident), with higher scores indicating higher CPR self-efficacy. While previous research has indicated Cronbach's alpha for this scale to be 0.82 [26], in this study, it was 0.91.

*CPR stress.* The CPR stress tool was developed by Cole, Slocumb [27], and it was translated and revised by Cha [28]. This tool consists of 20 items in total and five subscales, namely, confusing emotions, uncertainty, moral conflict, repression, and burden. The items are measured on a 5-point Likert scale from 1 (not at all painful) to 5 (very painful), where a higher score means higher stress when performing CPR. Here, Cronbach's alpha for this scale was 0.93, consistent with the previous research [28].

**Ethical consideration.** This study was performed after being granted formal approval by the hospital's institutional review board (CNUH-2021-194). Prior to the study's initiation, the researcher explained the purpose and processes of the study to the participants, all of whom provided written consent to participate. The participants were granted the freedom to

withdraw their participation for any reason without negative effects. Data confidentiality and anonymity were maintained throughout and after the study.

**Statistical analysis.**   SPSS Statistics for Windows, version 26 (IBM Corp., Armonk, NY, USA) was used to analyze the study data. Descriptive statistics were used to analyze the participants' demographic data, where means and standard deviations were calculated for continuous variables. The homogeneity of the participants' general characteristics was verified using the $\chi^2$ test, Fisher's exact test, or independent t-test analysis, as appropriate. The paired t-test and independent t-test were used to compare differences within and between outcomes. Given the differences in baseline stress, we also analyzed the data using analysis of covariance. The level of significance was set at $p \leq 0.05$.

## Results

### Homogeneity test of general characteristics and pretest outcomes

Each group had more than 30 participants, the skewness of the outcomes was −0.49 to 0.38, and kurtosis was −1.70 to 2.12, so the analysis was performed assuming normality of the data [29]. All general characteristics were homogeneous between the two groups. Additionally, ACLS knowledge and CPR self-efficacy were similar between the groups, but CPR stress was significantly higher in the IG than the CG (t = 2.03, $p$ = 0.044) (Table 1).

**Table 1.  Baseline characteristics and pretest outcomes of participants (N = 226).**

| Characteristics | Categories | Total (N = 226) n (%), M±SD | Intervention group (n = 117) N (%), M±SD | Comparison group (n = 109) N (%), M±SD | $\chi^2$ or t ($p$) |
|---|---|---|---|---|---|
| Age (year) | ≤25 | 33 (14.6) | 22 (9.7) | 11 (4.9) | 5.43 (0.066) |
| | >25 to ≤30 | 133 (58.8) | 70 (31.0) | 63 (27.9) | |
| | >30 to ≤52 | 60 (26.5) | 25 (11.1) | 35 (15.5) | |
| | Total | 29.8±5.6 | 29.2±5.1 | 30.6±6.0 | 1.89 (0.060) |
| Gender | Female | 218 (96.5) | 113 (50.0) | 105 (46.5) | 0.01 (1.000†) |
| Education | Bachelor's | 208 (92.0) | 114 (53.3) | 100 (46.7) | 3.637 (0.075†) |
| | Above Master's | 12 (5.3) | 3 (1.3) | 9 (4.0) | |
| Nursing experience (year) | ≤ 3 | 90 (38.8) | 53 (23.5) | 37 (16.4) | 3.80 (0.284) |
| | >3–≤ 5 | 59 (26.1) | 28 (12.4) | 31 (13.7) | |
| | >5–≤ 10 | 49 (19.0) | 22 (9.7) | 21 (9.3) | |
| | > 10 | 34 (15.0) | 14 (6.2) | 20 (8.8) | |
| | Total | 5.7±5.7 | 5.1±5.3 | 6.2±5.9 | 0.39 (0.164) |
| Experience of BLS education | Yes | 200 (88.5) | 102 (45.1) | 98 (43.4) | 0.41 (0.521) |
| Experience of ACLS education | Yes | 59 (26.1) | 32 (14.2) | 27 (11.9) | 0.20 (0.659) |
| Experience of simulation | Yes | 184 (81.4) | 99 (43.8) | 85 (37.6) | 1.64 (.200) |
| Experience of CPR | Yes | 184 (81.4) | 97 (42.9) | 87 (38.5) | 0.36 (0.551) |
| **Baseline variables** | **M±SD** | **M±SD** | **M±SD** | **t ($p$)** |
| ACLS Knowledge | 15.0±1.8 | 15.1±1.9 | 14.9±1.7 | 0.66 (0.512) |
| CPR Self-efficacy | 81.6±16.2 | 80.9±15.1 | 82.5±17.3 | 0.76 (0.450) |
| CPR Stress | 76.4±11.9 | 78.0±12.1 | 74.8±11.61 | 2.03 (0.044) |

*Note*. M±SD = mean±standard deviation; BLS = basic life support; ACLS = advanced cardiac life support; CPR = cardiopulmonary resuscitation;

† = Fisher's exact test

## Comparison of outcomes

The first hypothesis in this study was that the IG participants (those using the HTAS) would show greater ACLS performance than those in the CG. Levene's test showed unequal variances (F = 9.99, $p < 0.002$), so the degrees of freedom were adjusted from 224 to 190. Indeed, after the intervention, the total ACLS performance score of the IG was significantly higher than that of the CG (t = 50.80, $p < 0.001$) (Table 2). Among the subdomains of ACLS performance, BLS performance, CPR team activation, ACLS performance, endotracheal intubation assistance, and teamwork, were all significantly higher for the IG than for the CG (t = 32.68, $p < 0.001$; t = 33.01, $p < 0.001$; t = 42.73, $p < 0.001$; t = 33.50, $p < 0.001$; t = 36.71, $p < 0.001$, respectively).

The second hypothesis in this study posited that the IG participants would show greater ACLS knowledge and CPR self-efficacy than those in the CG. Before comparing the two groups, the pretest-posttest mean differences of each group were compared. For the IG, ACLS knowledge (t = 6.92, $p < 0.001$) and CPR self-efficacy (t = 6.97, $p < 0.001$) significantly increased when the pretest and posttest values were compared (Table 3). Similarly, the CG's ACLS knowledge (t = 4.85, $p < 0.001$) and CPR self-efficacy (t = 6.24, $p < 0.001$) increased. The CPR stress did not change significantly in either group.

**Table 2. Comparison of ACLS performance between the two groups (N = 226).**

| Variables | Group | Mean±SD | t (p) |
|---|---|---|---|
| Total ACLS performance | Intervention group | 150.5±5.1 | 50.8 (<0.001) |
| | Comparison group | 107.2±7.4 | |
| BLS performance | Intervention group | 41.6±2.2 | 32.68 (<0.001) |
| | Comparison group | 29.9±3.1 | |
| CPR team activation | Intervention group | 23.4±1.2 | 33.01 (<0.001) |
| | Comparison group | 16.6±1.8 | |
| ACLS performance | Intervention group | 33.1±1.4 | 42.73 (<0.001) |
| | Comparison group | 22.7±2.1 | |
| Endotracheal intubation assistance | Intervention group | 27.2±1.0 | 33.50 (<0.001) |
| | Comparison group | 18.9±2.4 | |
| Teamwork | Intervention group | 25.4±1.7 | 36.71 (<0.001) |
| | Comparison group | 19.1±0.8 | |

*Note*. ACLS = advanced cardiac life support; BLS = basic life support; CPR = cardiopulmonary resuscitation

**Table 3. Comparison within and between group about knowledge, self-efficacy, and stress (N = 226).**

| Variables | Group | Pre (a) Mean±SD | Post (b) Mean±SD | t (p)[£] | Mean Difference (b-a) Mean±SD | t (p)[‡] |
|---|---|---|---|---|---|---|
| ACLS Knowledge | Intervention group | 14.9±1.7 | 16.2±1.4 | 6.92 (<0.001) | 1.3±2.0 | 1.52 (0.130) |
| | Comparison group | 15.1±1.9 | 16.0±1.6 | 4.85 (0.001) | 0.9±1.9 | |
| CPR Self-efficacy | Intervention group | 82.5±17.3 | 88.8±12.1 | 6.97 (<0.001) | 7.9±12.3 | -0.42 (0.673) |
| | Comparison group | 80.9±15.1 | 89.7±14.6 | 6.24 (<0.001) | 7.2±12.1 | |
| CPR Stress | Intervention group | 78.0±12.1 | 78.0±11.1 | 0.04 (0.967) | 0.0±11.2 | -0.88 (0.378) |
| | Comparison group | 74.8±11.6 | 76.1±13.4 | 1.29 (0.201) | 1.3±10.9 | |

*Note*. SD = standard deviation; ACLS = advanced cardiac life support; CPR = cardiopulmonary resuscitation;

[£]: Paired t-test within groups;

[‡]: Mean difference independent t-test between groups

For hypothesis testing, the mean differences in ACLS knowledge, CPR self-efficacy, and CPR stress were compared. Levene's test indicated that the variances of the mean differences in ACLS knowledge, CPR self-efficacy, and CPR stress were equal (F = 0.98, $p$ = 0.324; F = 0.20, $p$ = 0.658; F = 0.01, $p$ = 0.936, respectively). There was no significant difference between the two groups concerning ACLS knowledge (t = 1.52, $p$ = 0.130), CPR self-efficacy (t = -0.42, $p$ = 0.673), or CPR stress (t = -0.88, $p$ = 0.378). For CPR stress, after controlling the baseline values, that is, the covariate, no significant mean difference was found (F = 0.00, $p$ = 0.984).

## Discussion

Clinical nurses are often the first responders to IHCA events [4]; therefore, they must be trained to perform ACLS proficiently. As face-to-face education has become challenging due to the recent spread of COVID-19, in line with the changing times, here, a variety of CPR training materials—which traditionally would have focused on in-person simulation [6, 13]—was compiled to develop a non-face-to-face education program. Thus, the HTAS developed in this study is meaningful in that it is an attempt to ensure effective and varied CPR training for nurses through combining a variety of online content and field simulations. This study demonstrated that the HTAS improved nurses' ACLS performance.

Various methods have been applied in attempts to improve CPR performance. In the context of simulation training, methods such as booster practice, feedback, and computer-controlled interactive manikins have been tested [30]. Other education strategies have included videos, popular songs, virtual reality content, gamification, and augmented reality [14, 15, 31]. Based on previous review studies, it can be said that there is sufficient evidence regarding the effectiveness of simulations, songs, and videos among various CPR education methods [14, 15, 30, 31]. Thus, our research team sought an effective and efficient education method to suit the current COVID-19 situation, and as a result, a hybrid method using online videos and simulations was adopted. Moreover, based on research on team-based simulation CPR education [9], our method focused on establishing theoretical frameworks, deliberate practices, and debriefing.

In contrast to previous studies investigating methods to improve clinical nurses' team-based ACLS performance, we divided the education program into three stages and adopted online-based training. Basic ACLS education was provided at any time as an online video program in the first stage, team-based ACLS skills training was provided in the second stage, and team-based ACLS simulation education was provided in the third stage. The IG participants undergoing the HTAS training scored significantly higher in ACLS performance than the CG participants. This suggests that hybrid education had a more significant impact on ACLS performance improvement than basic ACLS education only, consistent with other studies that investigated CPR simulation education for nurses [32, 33] and nursing students [34]. Since the hybrid team-based CPR program was shown to improve the ACLS performance of participants in this study, the program can be recommended for active application as a simulation-based training program for clinical nurses.

In a meta-analysis investigating the effect of CPR simulation training, the effect size for knowledge was 1.05, while that for product skill was 1.92, suggesting that simulation education was more effective at improving skills [30]. Specifically, the effect of the team-based CPR simulation method on skill improvement was clear, while improvement in knowledge was not found. Another review found that there was no association between skill improvement and knowledge in CPR training using manikins and that there was weak evidence for knowledge improvement, consequently calling for further studies [15]. However, in a study

that investigated simulation-based CPR training for hospital nurses with a control group, the nurses' knowledge improved [32]. In this study, ACLS knowledge significantly increased in both groups after the intervention, with no significant difference between the two groups. This result may have arisen because the online education program developed for ACLS knowledge acquisition in the first stage was equally applied to both groups. In other words, we do not believe that the team-based simulation training provided after the first stage contributed to the IG participants' knowledge improvement. This can be interpreted as aligning with the findings of previous review studies, which have suggested team-based CPR simulation education to be helpful in terms of skills and performance training but weak at improving knowledge [15, 30].

Likewise, the participants' self-efficacy significantly increased in both the IG and CG after the intervention, but there was no significant difference between the two groups in this regard. In a study comparing the effects of traditional simulation and rapid cycle deliberate practice simulation [22], confidence improved after the intervention, and there was no significant difference according to the type of education, similar to the results of this study, especially those of the posttest. For ACLS performance measurements in this study, after online learning, the CG was taken through simulation training. For this group, ACLS training was provided merely for evaluating CPR performance, so important factors, such as debriefing, feedback, and facilitation, were not included. However, since they underwent ACLS training, their CPR performance may have resulted in improved self-efficacy. When planning future interventional studies, it is necessary to design the follow-up period more precisely to be sure of this effect.

There were no significant mean differences in either the IG or CG in terms of CPR stress in this study before and after the intervention. The groups differed significantly from one another in terms of pretest CPR stress. In the baseline data of this study, the proportion of junior nurses with less than 3 years of nursing experience in the IG was higher than that of the CG. There was no significant difference in the average years of nursing experience, but the IG had a nonsignificant trend toward less experience. Therefore, differences in nursing experience may have led to differences in pretest stress scores. From another perspective, the CG underwent ACLS training only for the purpose of evaluating CPR performance; thus, important simulation elements—such as debriefing and feedback—were not included, which means the stress experienced by CG participants could have increased. Due to the nature of the simulation, the participants may feel the same anxiety and tension as in real situations and, therefore, experience stress [35]. In the IG, CPR stress may have been somewhat reduced by the positive feedback received in debriefings from the faculty, while in the CG, undergoing ACLS simulation without these important factors may have increased their CPR stress. This result contradicts the importance of debriefing after simulation training. To reduce CPR-related stress among clinical nurses, simulation training, including debriefing and feedback, should be developed.

This study had some limitations. First, it was conducted on clinical nurses at general tertiary hospitals located in one region, and participants were selected through convenience sampling. Therefore, caution should be taken when generalizing the results of the study. The effect of such education could be confirmed more clearly in future studies if a randomized controlled trial approach with multi-institutional participation is adopted. Second, this study was a one-time education course conducted over a 1-week period; thus, the long-term effects of the program could not be measured. Further studies should verify the continuity of the effect of the HTAS intervention. Additionally, since reinforcement through repetition training is important in CPR training [12], the online content of this study can be utilized for repetitive training.

## Conclusion

In this study, a hybrid team-based ACLS education program, including online content, was developed for nurses. After undergoing training based on the developed program, the IG nurses' ACLS performance improved compared to that of the CG, and when comparing the IG's outcomes before and after the intervention, their ACLS knowledge and CPR self-efficacy showed improvements. Since the hybrid team-based ACLS training was effective at improving ACLS performance, future research on the efficiency of team-based simulation, incorporating more diverse non-face-to-face content, is warranted. If various online CPR educational materials are developed, it may be possible to maintain the quality of education and enable repeated and continuous education.

## Supporting information

**S1 Fig. The hospital education website and uploaded lectures.**
(TIF)

## Acknowledgments

We sincerely thank Sun Hee Seon and Jin Young Lee for their assistance in data collection. The authors express their heartfelt appreciation to the participants and thank them for their valuable time and support.

## Author Contributions

**Conceptualization:** Hye Won Jeong, Sun-Hee Moon.

**Data curation:** Hye Won Jeong, Na Ru Kang, Eun Jeong Choi, Shin Hye Ahn.

**Funding acquisition:** Sun-Hee Moon.

**Investigation:** Hye Won Jeong, Na Ru Kang, Eun Jeong Choi, Shin Hye Ahn.

**Methodology:** Sun-Hee Moon.

**Resources:** Deok Ju, Ae Kyong Lee, Jung A Lee, Na Ru Kang, Eun Jeong Choi, Shin Hye Ahn, Sun-Hee Moon.

**Software:** Hye Won Jeong.

**Supervision:** Deok Ju, Ae Kyong Lee, Jung A Lee, Sun-Hee Moon.

**Visualization:** Hye Won Jeong.

**Writing – original draft:** Hye Won Jeong, Sun-Hee Moon.

**Writing – review & editing:** Sun-Hee Moon.

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
