## [Decision Letter · Decision Letter 0]

6 Sep 2022

PONE-D-22-04155Effect of a Hybrid Team-based Advanced Cardiopulmonary Life Support Simulation Program for Clinical NursesPLOS ONE

Dear Dr. Moon,

Thank you for submitting your manuscript to PLOS ONE. After careful consideration, we feel that it has merit but does not fully meet PLOS ONE’s publication criteria as it currently stands. Therefore, we invite you to submit a revised version of the manuscript that addresses the points raised during the review process.

While the content of your paper is valuable and addressing an interesting and important topic, the research lacks motivation in development of hypothesis. the authors can add little more to support in development of the hypothesis. Make sure that hypothesis are framed in context of the population not to sample (participants). In the result section, it is advised to add levene's test of homogeneity equality of  variances in groups and test of normality as required for application to t test. 

We look forward to receiving your revised manuscript.

Kind regards,

Prabhat Mittal, Ph.D.

Academic Editor

PLOS ONE

Journal Requirements:

“-The 10th authors of this study, Sun-hee Moon, received the fund.

-This work was supported by the National Research Foundation of Korea (NRF) grant funded by the Korea government (MSIT) (No. NRF- 2020R1I1A3063639).”

Reviewers' comments:

Reviewer's Responses to Questions

**Comments to the Author**

1. Is the manuscript technically sound, and do the data support the conclusions?

Reviewer #1: Partly

Reviewer #2: Yes

2. Has the statistical analysis been performed appropriately and rigorously? 

Reviewer #1: Yes

Reviewer #2: Yes

3. Have the authors made all data underlying the findings in their manuscript fully available?

Reviewer #1: Yes

Reviewer #2: Yes

4. Is the manuscript presented in an intelligible fashion and written in standard English?

Reviewer #1: No

Reviewer #2: No

5. Review Comments to the Author

Reviewer #1: General Comments:

Language needs editing.

Specific comments:

Abstract:

- It should be detailed with all subtitles "Background, methods, results, conclusion"

- Add detailed data about the participants.

- The conclusion should be precise. Add the future directions.

Introduction:

- This section does not cover all the elements of the study.

- Define "a Hybrid Team-based CPR" in detail.

- Explain the measured variables.

- The significance of the study needs more details.

- Add a clear hypothesis.

Methods:

- The study design, ethics, and setting are not clear.

- How and who administrates the data collection?

- How did you achieve the validity and reliability of the outcome measures?

- For statistical analysis, explain all methods used in detail and add the software used.

- Please, re-frame the components (SPICES) for methods

i. Study design, setting, sample size

ii. Participant

iii. Intervention/issue of interest (exposure)

iv. Comparison

v. Ethics and endpoint

vi. Statistical analysis

- What were the eligibility criteria for participants?

- Mention the settings and locations where the data were collected.

- Provide sufficient details of interventions of each group to allow replication.

- Define pre-specified primary and secondary outcome measures.

- Explain with reasons for any changes to study outcomes after the study commenced.

- How was the sample size determined?

- What was the method used to generate the random allocation sequence?

- Explain the type of randomization.

- Was there any restriction like blocking and block size?

- What kind of mechanism was used to implement the random allocation sequence?

- Were any steps taken to conceal the sequence?

- Who generated the random allocation sequence?

- Who enrolled participants?

- Who assigned participants?

- How was blinding addressed?

Results:

- Results need to provide answers to the questions raised/researchable problem

- Results need to follow ABC (accuracy, brevity, clarity)

- Kindly frame it along with the following elements of results

i. Text to tell the story

ii. Tables to summarize the evidence

iii. Figures to highlight the main findings

- Kindly provide dates defining the periods of recruitment and follow-up.

- This section needs to be put in the line with objectives.

- Explain Recruitment and Baseline data.

- Numbers analyzed need to be described well, especially in the column of "t and P values".

- Outcomes and estimation need to be explained well.

- Ancillary analyses and harms need to be addressed.

Discussion:

- Introductory paragraph should include the main findings of the study.

- This section needs to be put in the line with objectives and hypotheses.

- Explain the strengths and implications of the study in detail.

- The main limitation of the study design is not demonstrated.

Conclusion:

- The conclusion should be precise. Add the future directions.

Reviewer #2: The title of the manuscript is novel and appropriately justified with methods and material. The findings and conclusions are properly matched with the objectives of the study. Some part of the manuscript needs to be rephrased as per the standard format.

6. PLOS authors have the option to publish the peer review history of their article (what does this mean?). If published, this will include your full peer review and any attached files.

Reviewer #1: **Yes: **Walid Kamal Abdelbasset

Reviewer #2: No

---

## [Author Response · Author response to Decision Letter 0]

16 Nov 2022

My co-authors and I sincerely appreciate the reviewers’ feedback. We have revised the manuscript to reflect the reviewers’ opinions as much as possible. 

Details are attached as a file.

---

## [Editor Report · Decision Letter 1]

18 Nov 2022

Effect of a Hybrid Team-based Advanced Cardiopulmonary Life Support Simulation Program for Clinical Nurses

PONE-D-22-04155R1

Dear Dr. Moon,

We’re pleased to inform you that your manuscript has been judged scientifically suitable for publication and will be formally accepted for publication once it meets all outstanding technical requirements.

Kind regards,

Prabhat Mittal, Ph.D.

Academic Editor

PLOS ONE
---

## [Editor Report · Acceptance letter]

7 Dec 2022

PONE-D-22-04155R1 

Effect of a Hybrid Team-based Advanced Cardiopulmonary Life Support Simulation Program for Clinical Nurses 

Dear Dr. Moon:

I'm pleased to inform you that your manuscript has been deemed suitable for publication in PLOS ONE. Congratulations! Your manuscript is now with our production department. 

Kind regards, 

on behalf of

Dr. Prabhat Mittal 

Academic Editor

PLOS ONE